# Genetic Variability of Bovine Leukemia Virus: Evidence of Dual Infection, Recombination and Quasi-Species

**DOI:** 10.3390/pathogens13020178

**Published:** 2024-02-15

**Authors:** Aneta Pluta, Marzena Rola-Łuszczak, Federico G. Hoffmann, Irina Donnik, Maxim Petropavlovskiy, Jacek Kuźmak

**Affiliations:** 1Department of Biochemistry, National Veterinary Research Institute, 24-100 Puławy, Poland; mrolka@piwet.pulawy.pl (M.R.-Ł.); jkuzmak@piwet.pulawy.pl (J.K.); 2Department of Biochemistry, Molecular Biology, Entomology and Plant Pathology, Mississippi State University, Starkville, MS 39762, USA; fgh19@msstate.edu; 3Institute for Genomics, Biocomputing and Biotechnology, Mississippi State University, Starkville, MS 39762, USA; 4Ural State Agrarian University, Ekaterinburg 620075, Russia; ktqrjp7@yandex.ru; 5Ural Federal Agrarian Scientific Research Centre of the Ural Branch of the Russian Academy of Sciences, Ekaterinburg 620049, Russia; petropavlovsky_m@mail.ru

**Keywords:** BLV, *Deltaretrovirus*, dual infection, recombination, quasi-species

## Abstract

We have characterized the intrahost genetic variation in the bovine leukemia virus (BLV) by examining 16 BLV isolates originating from the Western Siberia–Tyumen and South Ural–Chelyabinsk regions of Russia. Our research focused on determining the genetic composition of an 804 bp fragment of the BLV *env* gene, encoding for the entire gp51 protein. The results provide the first indication of the quasi-species genetic nature of BLV infection and its relevance for genome-level variation. Furthermore, this is the first phylogenetic evidence for the existence of a dual infection with BLV strains belonging to different genotypes within the same host: G4 and G7. We identified eight cases of recombination between these two BLV genotypes. The detection of quasi-species with cases of dual infection and recombination indicated a higher potential of BLV for genetic variability at the intra-host level than was previously considered.

## 1. Introduction

Point mutations and recombination are two of the most important sources of genetic variation. Both mechanisms are prevalent among members of the *Retroviridae* family; point mutations act as the source of novel variants, which are then combined into novel haplotypes through recombination. The quasi-species concept, which is defined as the population of viruses that inhabit a single host, explicitly considers these mechanisms and can be applied to retroviruses [1]. When dual infection of one host by relatively different viral strains occurs, complex patterns of sequence diversity can appear [2]. The best example comes from viruses such as Human Immunodeficiency Virus-1 (HIV-1), a member of the genus *Lentivirus* in the *Retroviridae* family, which has an extremely high population diversity due to the high error rate of reverse transcriptase and long periods of infection [3].

In contrast to most other genera in *Retroviridae*, viruses in the genus *Deltaretrovirus*, which includes human and simian T-lymphotropic viruses (HTLV and STLV), are characterized by relatively low levels of genetic diversity. However, the quasi-species nature of some HTLV-1 infections and relatively high levels of variability of HTLV-1 sequences within a single viral strain have been reported [4,5]. Furthermore, co-infection with two different types of HTLV or STLV and recombination between different subtypes of HTLV-1 have been identified as well [6,7,8].

In contrast to HTLV and STLV, the occurrence of the quasi-species type of infection and natural recombination in the bovine leukemia virus (BLV) has never been documented. The BLV is the causal agent of enzootic bovine leukosis (EBL) and is also a member of *Deltaretrovirus*. Most of the infected cattle remain clinically asymptomatic although one-third of them develop persistent lymphocytosis which is characterized by the polyclonal expansion of infected B-lymphocytes, and a small percentage of the latter develop leukemia/lymphoma [9]. BLV is distributed worldwide and there are several regions where infections are highly prevalent, such as Argentina, USA, and Japan [10,11,12,13]. Based on phylogenetic studies of BLV isolates from all over the world, the BLV strains are currently classified into 12 different genotypes which largely reflect their geographical distribution [14,15]. In a previous study we reported that the most common BLV strains from Eastern Europe and Siberia belong to genotypes G4, G7 and the newly identified genotype G8 [15]. This study also revealed that in most regions of Russia genotypes G4 and G7 might circulate in animals from the same area and *env* gene sequences from some of these animals showed the existence of multiple polymorphic sites. This suggests the presence of a heterogeneous virus population within one host, as was pointed out for HTLV-1 and STLV-1 4. The previous research was based on analyses of a 444 bp fragment of the *env* gene. In the current study, we estimate the levels of nucleotide diversity in the portion of the *env* gene encoding for the whole gp51 glycoprotein of *env* gene from cattle infected with BLV from Russia. We demonstrated that a quasi-species type of infection, dual infection with the genotypes G4 and G7, and a recombination were presented under natural infection.

## 2. Materials and Methods

Blood samples were collected from 16 cattle that had tested positive for BLV as indicated by an agar gel immunodiffusion test (AGID). Eight of the samples (designated as 1S, 2S, 3S, 4S, 5S, 4T, 5T, 6T) came from the Tyumen region, in Western Siberia, and eight (designated as 1C, 7C, 8C, 9C, 10C, 2Z, 4Z, 5Z) originated from the Chelyabinsk region, in South Ural, both in Russia (Figure 1).

Genomic DNA was isolated from peripheral blood leukocytes (PBLs) (DNeasy Tissue Kit, Qiagen). The amplicons of *env*–gp51 (804 bp) were obtained through a two-step PCR using primer pairs P4796, P5791 and P4833, P5734 for the first and second round, respectively, under the following conditions: 95 °C 3 min, 95 °C 30 s, 62 °C 30 s (external primers) or 66 °C 30 s (internal primers), 72 °C 2 min for 35 cycles, and final extension for 10 min 72 °C. The pairs of primers used in this study are shown in Appendix A. The reaction consisted of 500 ng of template DNA, 1 × optimized DyNAzyme buffer, 200 μM each dNTPs, 0.5 μM each primer, 2 mM MgCl_2_ and 2 U/50 μL of DyNAzyme II DNA Polymerase (Thermo Scientific, Waltham, MA, USA). The DyNAzyme II DNA Polymerase mutation rate is 2.28 × 10^−5^ mutations per base pair per template duplication. In a two-step PCR protocol performed with this polymerase, each product molecule could contain an average of 1.28 errors (1.28 nucleotides) (see the legend of Appendix A for detailed calculations). In order to estimate the potential contribution of PCR errors to sequence variability, three independent PCR products for each DNA sample were sequenced. PCR fragments obtained from the isolates 1S and 4T were cloned into the pCR 2.1 vector with a TA cloning kit (Invitrogen, Waltham, MA, USA) and then 20 resulted clones were sequenced per each isolate. Sequence data were analyzed and aligned using the Geneious Alignment module within Geneious Pro 5.3 Software (Biomatters Ltd. Auckland, New Zealand).

Polymorphic and parsimony informative sites were computed using the Nei–Gojobori method as implemented in DnaSP 5.10.01 [16]. To assess selective pressures, the dN/dS ratio of each of the individual 20 clones 1S and 4T was calculated by using the function of the DnaSP 5.10.01. To estimate nucleotide diversity between strains from the same animal, analyses were conducted using the p-distance model in MEGA X [17,18]. To infer phylogenetic relationships of the collected BLV isolates, the sequences were aligned to a representative panel of 27 *env* gene sequences encoding the gp51 protein, which included all currently recognized BLV genotypes, 1 to 12, with a wide geographic distribution (Table 1). Phylogenetic analysis was performed using the Bayesian inference of phylogeny as implemented in Geneious Pro, and support for the nodes was evaluated with posterior probabilities.

To explore the possibility of recombination a pairwise homoplasy index (Phi) test was performed and phylogenetic networks were computed using the program SplitsTree4 [32]. The number and positions of the inferred recombination breakpoints and trees derived from the segmentation of the sequence alignment were calculated using the program PhyML_Multi, which is built upon the algorithmic structure of PhyML [33]. BIONJ trees were simultaneously constructed using the PhyML_Multi program.

## 3. Results

The PCR products of the expected size of 804 bp were successfully amplified and complete *env*–gp51 gene sequences were obtained from all 16 isolates. The sequence analysis revealed that for 14 isolates, the sequencing of three independent PCR products yielded almost identical data, with pairwise identities ranging from 99.9% (3 samples) to 100% (11 samples). The corresponding consensus sequences were deposited in GenBank and accession numbers were assigned as follows: 2S–5S (JF720350–JF720353); 5T (HM563754); 6T (HM563783); 1C, 7C–10C (JF720354–JF720358); 2Z (JQ320302); 4Z (HM563779); and 5Z (HM563780). However, in the case of the 1S and 4T isolates, the sequencing of the three PCR products revealed the presence of multiple peaks in the electropherograms, suggesting the presence of a heterogenous population of the virus in these hosts [34]. PCR products were then cloned, followed by the sequencing of 20 clones per isolate. The resulting sequences were submitted to GenBank with the accession numbers JQ353633–JQ353652 and JQ353653–JQ353672 for the isolates from 1S and 4T, respectively.

Sequence analysis using the Nei–Gojobori method identified the presence of 70 and 60 polymorphic sites with 22 and 27 parsimony informative sites for viral isolate 1S and 4T, respectively. The p-distance analysis showed that the mean genetic distance among these 20 clones was 2.0% (range: 0.1–4.5%) and 2.1% (range: 0–4.7%) for isolates 1S and 4T, respectively, as seen in Figure 2 and Figure 3.

The ratio of *synonymous* (dS) and *nonsynonymous* (dN) substitutions within *env* sequences representing 20 clones of 1S and 20 clones of 4T isolates was calculated (Appendix A). The dN/dS ratios were drawn over the midpoint window position (window length 20, step size 10) from the whole coding region. The following regions with putative positive selection sites were identified: 61–80 nt, 111–130 nt, 201–220 nt, 251–270 nt, 331–350 nt, 761–780 nt in 1S isolate sequences and 131–150 nt, 251–270 nt, 351–370 nt for 4T isolate sequences, respectively. Seven codons located in these regions had dN/dS ratios >1 that identified them as major sites for the occurrence of positive selection. These were codons 23, 40, 88 in conformational epitopes G and H; 50 in CD8 T-cell epitope; 111, 118 in overlapping neutralization domain 2, CD8+ T-cell epitope and Zinc-binding peptide; and 258 in linear epitope A (Appendix A).

The Bayesian phylogenetic tree using the entire set of sequences was constructed to cluster BLV isolates from the Tyumen and Chelyabinsk regions. The analysis of the tree showed that fourteen isolates clustered clearly with either genotype G4 (6T, 2Z, 4Z, 5Z, 1C, 7C, 8C, 9C, 10C) and G7 (2S, 3S, 4S, 5S, 5T), (Figure 4).

When 40 sequences representing two isolates 1S and 4T were analyzed, inconsistent phylogenic assignment was noted, suggesting that they could come from cattle co-infected with viruses belonging to the genotypes G4 and G7. For the isolate 1S, eight clones were clustered with genotype 7 and twelve with genotype 4, while for isolate 4T, eleven and nine clones belonged to genotype 7 and 4, respectively (Figure 4). The structure of variation within the isolates 1S and 4T, with multiple independent clusters of sequences, indicated that the identified branches were composed of variants showing the quasi-species nature of BLV. These results clearly showed a high level of intra-host heterogeneity, possibly leading to the recombination events. To test this hypothesis, as a first step we employed the Phi test to analyze the 20 sequences of clones representative for each isolate 1S and 4T which were compared with 14 sequences used in this study and an additional 27 BLV reference sequences. The results of pairwise homoplasy index tests detected statistically significant evidence for recombination events occurring among sequences representing a heterogenic population of BLV virus both in isolate 1S and 4T with *p* values ranging from to 3.347 × 10^−6^ to 2.519 × 10^−3^ for the sequences from isolates 1S and 4T, respectively. To identify particular sequences with evidence of recombination, phylogenetic networks using NeighbourNet mode for the same set of sequences were estimated. As shown in Figure 5 and Figure 6, the sequences representing four clones of the isolate 1S (1S-c1, 1S-c2, 1S-c9 and 1S-c11) and four clones of the isolate 4T (4T-c1, 4T-c19, 4T-c20, 4T-c21) had positions in the network that were indicative of recombination.

Furthermore, the same set of sequences was subjected to breakpoint analysis using PhyML_Multi program. Two breakpoints were predicted for four sequences from isolate 1S at positions 240 bp and 542 bp (Appendix A). Accordingly, three separate BIONJ trees based on the different portions of the alignment clearly identified the recombinant sequences: 1S-c11 (G4/G4/G7), 1S-c9 (G7/G4/G4), 1S-c2 (G4/G7/G7) and 1S-c1(G7/G4/G7) (Appendix A). When sequences derived from the 4T isolate were analyzed, one breakpoint at position 400 bp of the alignment was predicted. Two BIONJ trees for each portion of the alignment revealed four recombinant sequences: 4T-c1 (G4/G7) and 4T-c19, 4T-c20, 4T-c21 (G7/G4) (Appendix A). To confirm the incongruence patterns observed in BIONJ trees, we verified these results with Bayesian phylogenies, and the results were similar, identifying the same eight sequences with recombination between genotypes 4 and 7 (Figure 7 and Figure 8). Genotype-specific nucleotides in the different fragments manually designated and individually confirming the breakpoints calculated using the Phylo-HMM algorithm, for each recombinant breakpoint, are shown in Appendix A. The final characteristics of the *env* sequences representative for 20 clones of each isolate, 1S and 4T, respectively, are shown in Table 2.

To rule out the possibility that sequences from a multi-sample population, amplified according to our two-step PCR protocol, could generate artificial chimeras, we performed a separate study. A mixture containing equal amounts of two genomic DNA samples (each with a copy number of 3 per 10^3^ cells), Goldap/22 and Szczytno/101 isolated from PBL from Polish BLV-infected cattle representing genotypes G4 and G7, respectively, was amplified (Appendix A). The PCR product G4 + G7 was cloned into a plasmid cloning vector and 20 clones were randomly selected to prepare plasmid DNA, then both strands of the plasmid DNA were sequenced (sequence data are shown in Appendix A). The phylogenetic analysis of 20 sequences representing the population with two templates revealed eight clones clustered with the G7 genotype and 12 clones with the G4 genotype (Appendix A). Detailed sequence alignment analysis of the clones compared to the Goldap/22 and Szczytno/101 sequences did not show the presence of recombinant forms between these genotypes or the presence of new genotypes.

## 4. Discussion

In this study, we estimate genetic diversity and phylogenetic relationships for a sample of BLV isolates from 16 cattle specimens from the southwestern part of the West Siberian Plain. Our analyses indicate that 14 of the infected cattle were infected by a highly homogeneous viral population with intrahost mean pairwise distances ranging from 0% to 0.1%, and 9 of genotype G4 and 5 of genotype G7. By contrast, the other two specimens harbored a much more diverse viral population, with intrahost mean pairwise distances of 2.0% and 2.1%. These results confirm the findings of our previous study indicating that the G4 and G7 genotypes predominantly circulate in BLV-infected cattle in this part of the world [15]. However, an unexpected result of this analysis was our inability to determine the classification of BLV provirus isolates infecting two individuals, named 1S and 4T. PCR cloning and detailed sequence analysis revealed the presence of a dual infection in these cattle by proviruses belonging to both the G4 and G7 genotypes. These dually infected animals were found in herds, where the BLV seroprevalence was nearly 75% and the infections with both genotypical isolates coexisted on one farm. The existence of mixed infections with multiple BLV strains was described under field conditions by Asfaw et al. but their study was limited to RFLP analysis [35], and by Camargos et al., who performed in vitro virus amplification [36]. Our results show the presence of dual BLV infection with viruses belonging to genotypes G4 and G7. It is well known that the direct contact between animals is a main route of virus transmission under natural conditions [37]. Therefore, it can be assumed that virus transmission in these herds occurred via this route, and as the animals were naturally exposed to BLV, it was difficult to determine if dual infection occurred sequentially as superinfection or simultaneously as co-infection. With regard to the present research, the issue of the occurrence of a superinfection or co-infection event is a critical question. Through the analogy of the model of cattle immunized with an attenuated virus and exposed to a circulating wild-type virus with high prevalence in the animal population, it can be assumed that it is rather impossible to infect an already infected cow through another BLV due to the strong host immune response [38]. Thus, superinfection seems illogical, as even cows immunized with the attenuated strain did not become infected with BLV wild-type from other cows kept with them for over 3 years [38]. Although the mechanism for the development of the dual infection, in our case, cannot be elucidated in detail, we suppose that it was possibly an accidental co-infection with dual pathogen genotypes. We also speculate that another possible mode of transmission of the virus was most likely blood-sucking midges. Western Siberia is rich in river systems which creates favorable conditions for insect reproduction [39,40,41].

Genetic stability is a typical feature for viruses belonging to the Deltaretrovirus genus and distinguishes them from other known retroviruses. Despite the relatively low level of variability between strains, some studies have shown an intrastrain variability in the *env* gene, as was reported for HTLV-1 [4]. However, this knowledge has been neglected in the study of the genetic variability of BLV. Our results are the first to report the presence in two animals of 1S and 4T, two clearly distinct proviral populations, showing quasi-species nature. We speculate that this event corresponded to a very early period after infection when the samples were collected, and then the active replication of virus had been manifested by existence heterogeneities in the intra-individual virus population. During primary infection, when an infected cell with the integrated BLV genome is transmitted from an infected animal, the virus actively replicates and infects a number of new cells (the RT-dependent replicative cycle). This phase of BLV infection is limited in time and lasts from 1 to 4 weeks after infection [42]. The early BLV replication is an RT-dependent process and that would be why the BLV genetic variability could be the highest during primary stage of disease. Additional support for this hypothesis comes from a study by Pomier et al. [43] study carried out on experimentally BLV infected sheep, showing that during the primary infection of a new host substitutions generated by RT lead to a mutation load accounting for 69% of the provirus genetic variability on the intra-host level. Watanabe et al. [44] postulated that the BLV superinfection process with defective BLV virus had its place during the early stage of infection. In addition, a second argument, providing that 1S and 4T animals were in early step of viral infection, was the relatively low proviral load of 0.35 and 11.3 copy numbers per 10^3^ cells found in these animals, respectively. In comparison to others studies, similarly low levels of the proviral load (of approximately 1 copy per 10^3^ cells PBMCs) were observed in cows 2 weeks post inoculation with a cloned BLV provirus (strain 344) [45]. This illustrates that these two animals did not show any progression of BLV-induced disease during the BLV infection.

The genetic divergences between the sequences of the clones representing each isolate from these animals were up to 5% and these values were similar to those noted for the distances between sequences representative for the twelve known BLV genotypes [30,46]. In light of these findings, we hypothesize that the broader range of genetic variability observed for the sequences of isolates 1S and 4T could be related to viral persistence in the infected host. In addition, the increased genetic diversity among the isolates could trigger differences in the genotypic and phenotypic features of the replication-competent proviruses and a broader range of infectivity. For this reason, long-term field investigations regarding the infected cattle and infected herds from which the samples originated are required.

The ratio of nonsynonymous sites (dN) to synonymous sites (dS) within functional domains and epitopes was calculated for *env* sequences representing 20 clones of 1S and 20 clones of 4T isolates (Appendix A). A significantly high dn/ds value was noted for the G epitope, H epitope, CD8 T-cell epitope and Zinc-binding peptide, indicating a positive selection for these fragments of *env* gene [47]. Several studies have reported that the mutations within the G and H epitopes of the *env* gene can lead to structural changes underlying viral escape from immune surveillance [21,23,27,48]. Similarly, the role of CD8 T-cell epitopes in the suppression of the progression of the disease caused by the virus correlated with the proviral load in BLV-infected cattle [49,50]. Therefore, we suppose that BLV variants identified here as quasi-species could be undergoing positive pressure during field infection leading to the development of an adopted form which can escape from immune surveillance.

The most interesting finding of our study was the discovery of BLV recombinants among dually infected cattle. Based on the results of phylogenetic network analysis, we assumed that the recombinant forms were present in PCR clones, generated from both 1S and 4T animals. The recombination occurred between genotype G4 and G7 and was accompanied by breakpoints distributed in three domains within the gp51 coding region, CD4+ T-cell epitope (240 bp), ND3 (542 bp) and CD8+ T-cell epitope (400 bp). The possibility of infection of one cell by at least two different copies of BLV was described inter alia by Watanabe et al. and Gutierrez et al. [51]. Based on the well-described recombination for HIV, recombination can occur if two viruses from the same species infect the same host cell and viral genome templates are exchanged during replication. Accordingly, for BLV, this could occur if the infection of the same B cell by the two different genotypes G4 and G7 leads to the production of virions that pack an RNA molecule of both genotypes (Figure 9A,B).

When such virions had infected subsequent target cells, they could produce a mosaic genome by the exchange of genes or gene fragments due to template switching in RT that resulted in recombinant viruses such as 1S-c11, 1S-c9, 1S-c2, 1S-c1, 4T-c1, 4T-c19, 4T-c20 and 4T-c21, which were recognized as circulating recombinant forms (Figure 7B and Figure 8B). Up to now, BLV genotypes G6 and G11 both in the same region have been reported in China [52]; the G1, G5, and G7 were present in Mongolia [53]; G1 and G3 were found in Korea [54] and G1, G3, and G5 were circulating in Japan [35,55]. However, there were not identified isolates that inherited the variation sites of two or more genotypes together. Theoretically, there is a potential for recombination between the isolates of different genotypes for both BLV isolates coexisting in the same farm, in the same animal. The identified recombination between G4 and G7 is a new phenomenon for BLV, and such an event may yield new genotypes or sub-genotypes, which is an interesting theme for further research. The major point of our study is to highlight the need for a broader study of the intrahost variability of BLV. Therefore, future research should focus on experimentally infected animals with different genotypes, BLV isolates, and a longitudinal study regarding the level of antibodies, proviral load, time of appearing quasi-species and recombination forms and also an acceleration of disease progression. In such studies, the use of NGS and inverse PCR (IPCR) should undoubtedly play a leading role as the optimal tool for studying intrahost genetic variability. Unfortunately, in the studies presented here, we had a limited amount of genetic material at our disposal which precluded the use of NGS. Furthermore, it is known that for DNA samples with the presence of provirus below 500 copies per 10^5^ cells, NGS and IPCR is of limited use, so in this case this technology was not applied due to the provirus load in the samples being too low.

## 5. Conclusions

These results are the first to report a dual infection with two different genotypes of BLV in the same host, which was associated with the detection of diverse recombinant forms and viral quasi-species. In BLV infected cattle, this phenomenon is probably only accidental in nature. Most likely, recombination was the result of RT activity that has to be preceded by the infection of the same cell by different BLV strains. The presence of multiple variant genomes (quasi-species) in infected cows was probably recorded during early co-infection with viruses. As a consequence, these results shed new light on BLV variability, particularly in the context of the intrahost variation, which can be higher than it was previously thought. For further study, the degree of intrahost genetic variability and cross-sectional samples of BLV provirus produced in one animal at one time point should be investigated.

## Figures and Tables

**Figure 1 pathogens-13-00178-f001:**
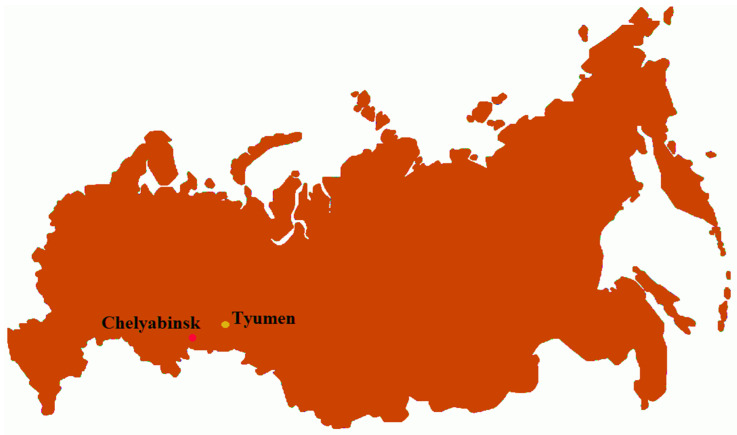
Map with locations of sample collection. Maps showing the location of the study area in the Tyumen (Koktiul) and Chelyabinsk (Koelga) regions of the West Siberian Plain (the Russian Federation).

**Figure 2 pathogens-13-00178-f002:**
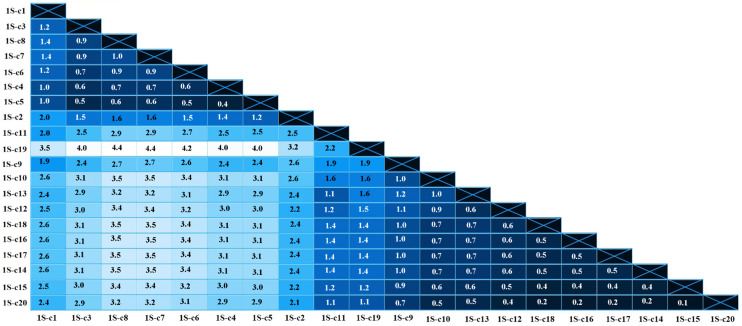
Heatmap of pairwise genetic distances (%) calculated using the alignment of 20 clones from 1S isolate. The numbers show the proportion of nucleotide substitutions in the 804 bp fragment of *env* gene.

**Figure 3 pathogens-13-00178-f003:**
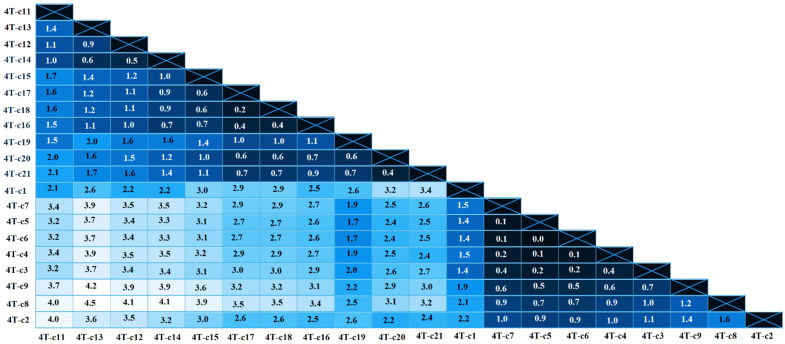
Heatmap of pairwise genetic distances (%) calculated using the alignment of 20 clones from 4T isolate. The numbers show the proportion of nucleotide substitutions in the 804 bp fragment of *env* gene.

**Figure 4 pathogens-13-00178-f004:**
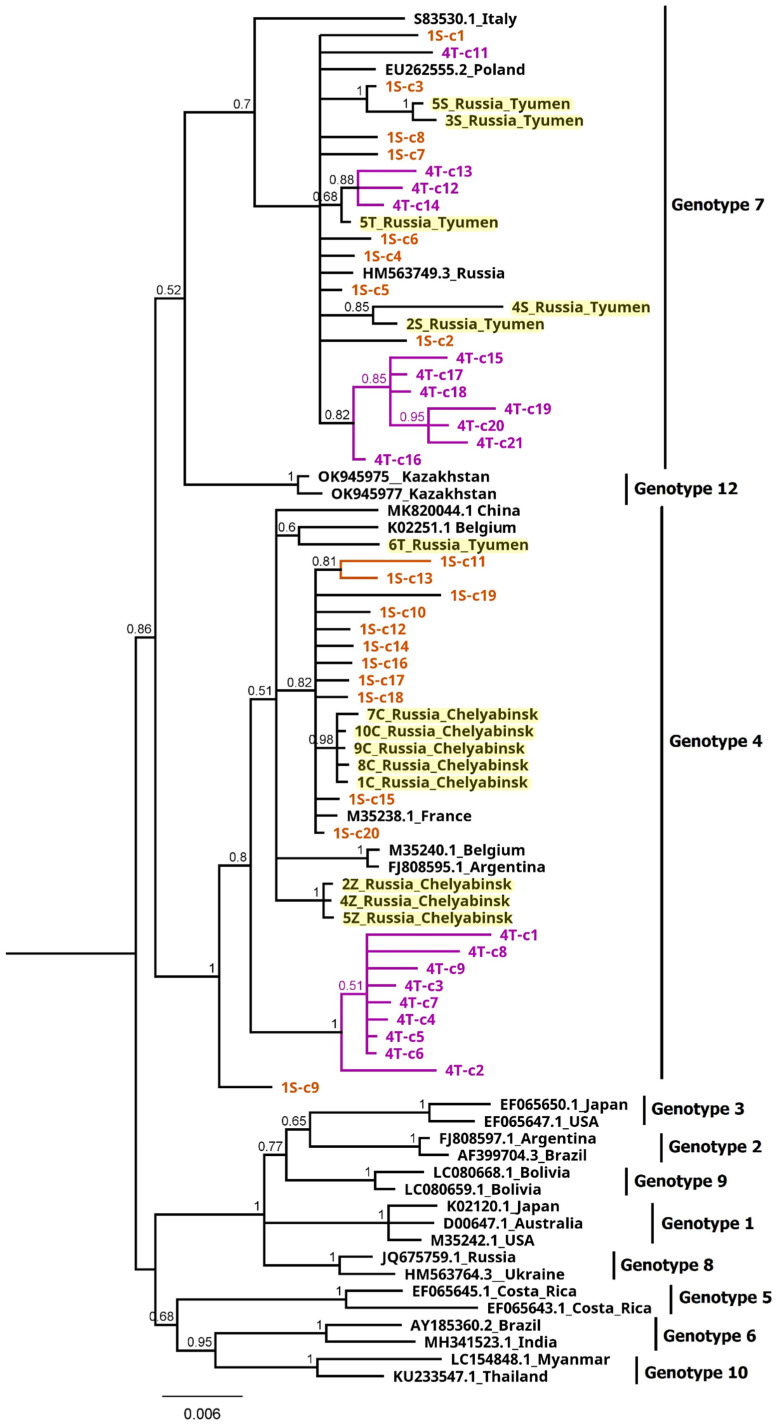
Bayesian phylogenetic tree based on 804 bp of *env* gene sequences of BLV isolates. The tree was midpoint rooted, indicated in the right site of tree by G1–G10 and G12. Numbers on nodes indicate posterior probabilities. The 83 sequences used in this analysis derived from 20 clones from each 1S (orange) and 4T (violet) isolate, 14 remaining isolates (shaded in yellow) and 27 reference data (black). The G11 genotype was not included in the phylogenetic analysis due to the lack of full gp51 length sequences existing in available databases.

**Figure 5 pathogens-13-00178-f005:**
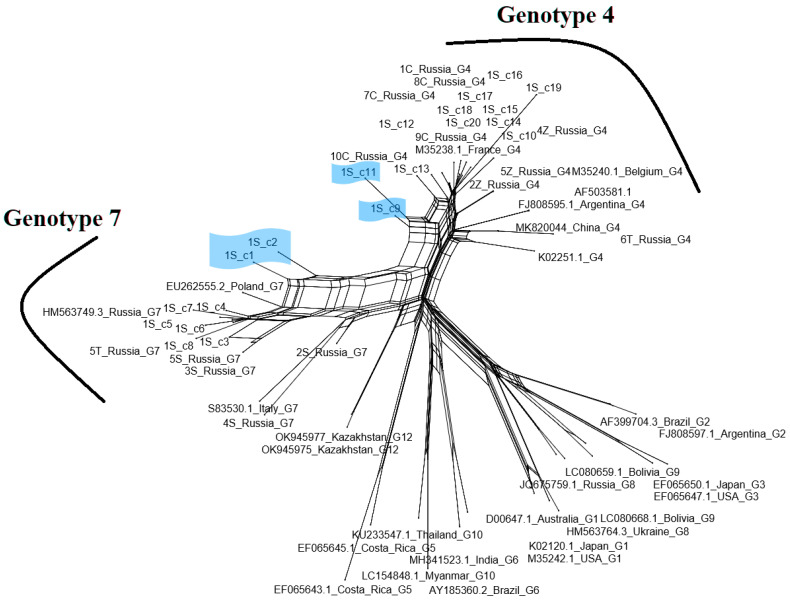
Phylogenetic network inferred using the NeighborNet model for 20 sequences representative for clones of the isolate 1S, analyzed with 14 sequences from the Tyumen and Chelyabinsk regions and 27 reference BLV sequences, available from GenBank. Sequences classified as genotypes 4 and 7 are ticked on the edge of the network. The putative recombinants 1S-c1, 1S-c2, 1S-c9 and 1S-c11 are shaded in blue. The fit index for the split network was 98.3%. A fit index above 90% is considered as robust.

**Figure 6 pathogens-13-00178-f006:**
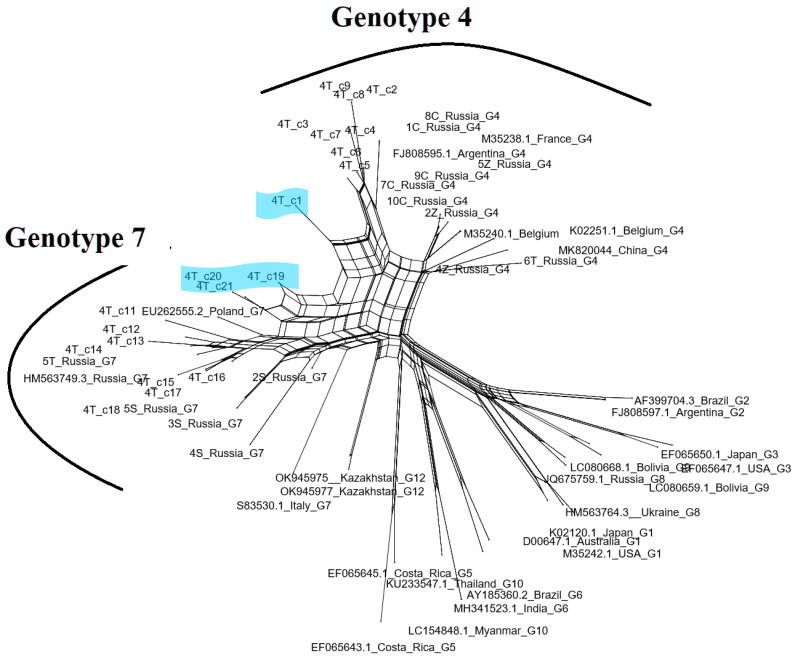
Phylogenetic network inferred using the NeighborNet model for 20 sequences representative for clones of the isolate 4T, analyzed with 14 sequences from the Tyumen and Chelyabinsk regions and 27 reference BLV sequences, available from GenBank. Sequences classified as genotypes 4 and 7 are ticked on the edge of the network. The putative recombinants 4T-c1, 4T-c19, 4T-c20, 4T-c21 are shaded in blue. The fit index for the split network was 96.5%. A fit index above 90% is considered as robust.

**Figure 7 pathogens-13-00178-f007:**
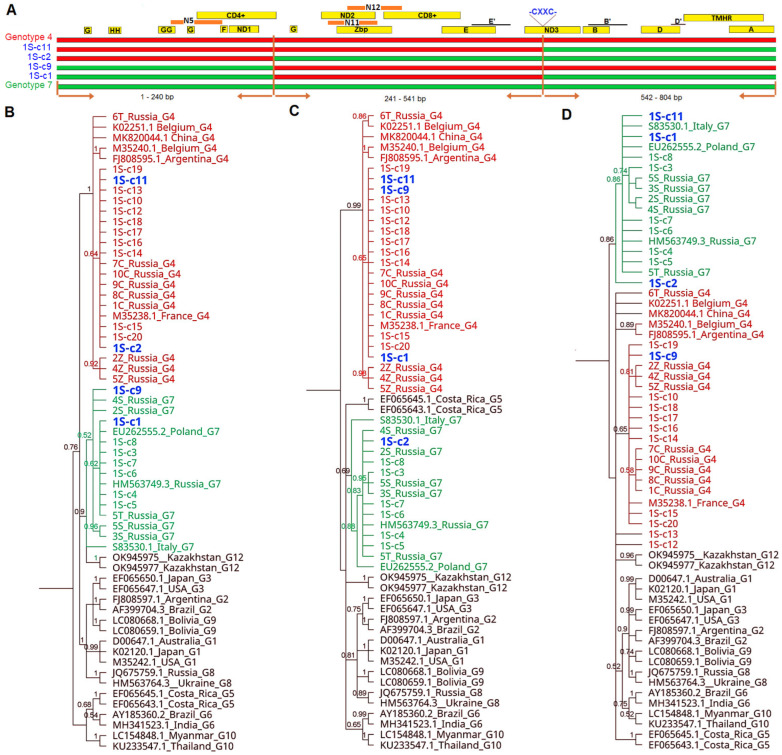
Bayesian phylogenetic trees from clone sequences from isolate 1S. (**A**) Color representation of recombinant sequences with portions similar to genotype G4 in red and portions similar to genotype G7 in green. Horizontal brown lines indicate the location of inferred recombination breakpoints. Labeled rectangles in the upper part of figure (yellow, orange) refer to the coding sequences of antigenic determinants. Epitopes A, B, B′, D, D′, E, E′ (linear), F, G, H (conformational), ND1,2,3–neutralization domain, CD4+, CD8+, N5, N11 and N12–T cell epitopes, Zbp-Zinc-binding peptide, GYDP strong turn, THMR–transmembrane hydrophobic region, CXXC sequence in disulfide bond. (**B**–**D**) trees are based on three separate non-recombining regions of *env* gene, identified via the Phylo-HMM algorithm and referred to as the 5′ terminus 1–240 bp, 241–542 bp, and 543–804 bp regions, respectively. Label names are shown in black for G1–G3, G5, G6, G8–G10 and G12 genotype reference sequences, green for genotype 7 sequences, red for genotype 4 sequences and blue for putative recombinant sequences.

**Figure 8 pathogens-13-00178-f008:**
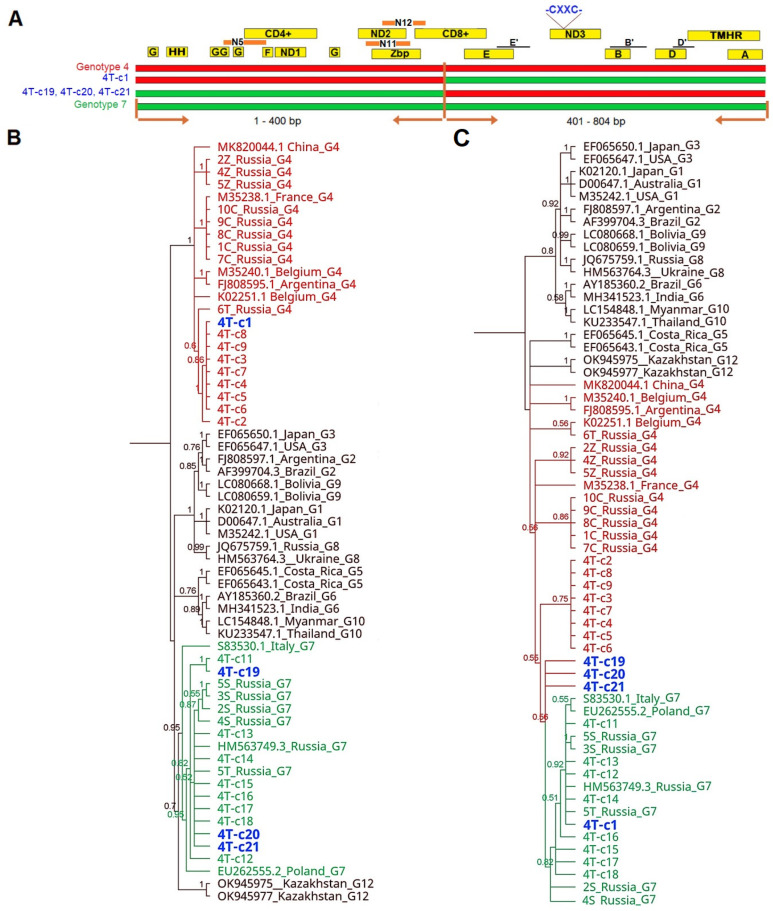
Bayesian phylogenetic trees from clone sequences from isolate 4T. (**A**) Color representation of recombinant sequences with portions similar to genotype G4 in red and portions similar to genotype G7 in green. Horizontal brown lines indicate the location of inferred recombination breakpoints. Labeled rectangles in the upper part of figure (yellow, orange) refer to the coding sequences of antigenic determinants. Epitopes A, B, B′, D, D′, E, E′ (linear), F, G, H (conformational), ND1,2,3–neutralization domain, CD4+, CD8+, N5, N11 and N12–T cell epitopes, Zbp-Zinc-binding peptide, GYDP strong turn, THMR–transmembrane hydrophobic region, CXXC sequence in disulfide bond. (**B**,**C**) trees are based on two separate non-recombining regions of the *env* gene, identified through the Phylo-HMM algorithm and referred to as 5′ terminus 1–400 bp and 401–804 bp regions, respectively. Label names are shown in black for G1–G3, G5, G6, G8–G10 and G12 genotype reference sequences, green for genotype 7 sequences, red for genotype 4 sequences and blue for putative recombinant sequences.

**Figure 9 pathogens-13-00178-f009:**
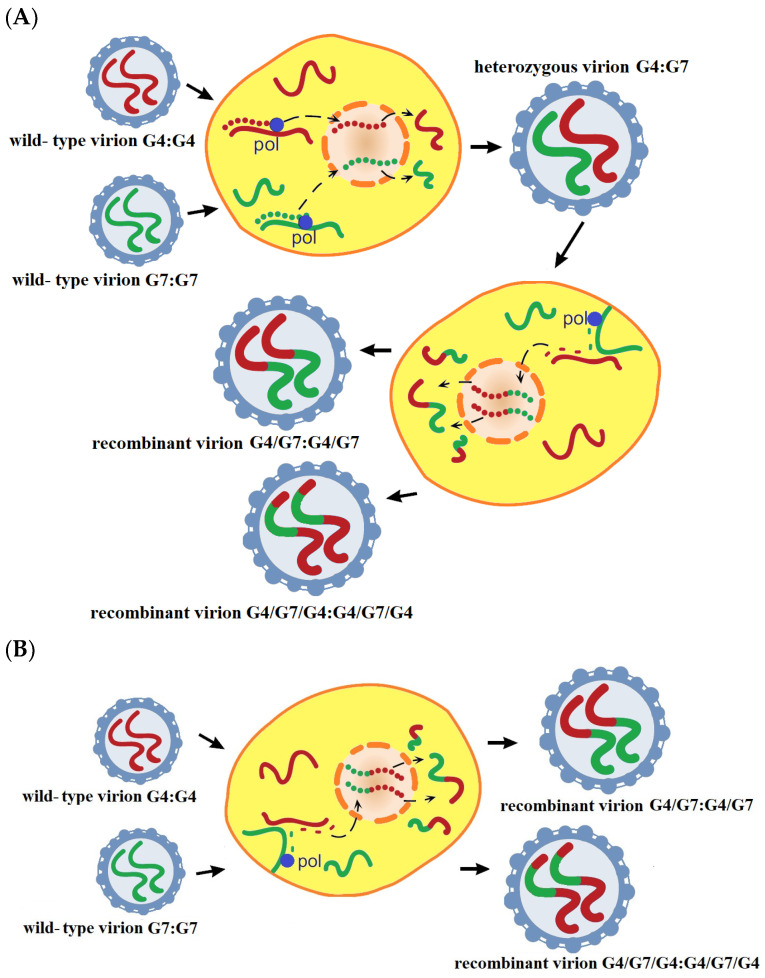
Illustration of the putative recombination in BLV. The scheme illustrates the possible mechanisms of recombination as it might occur in BLV (**A**,**B**). BLV is a diploid retrovirus for which, when a host cell was simultaneously infected with two strains of BLV (belonged to G4:G4 and G7:G7) and hence harbored two different proviruses, the RNA transcript from each of the BLV proviruses could be incorporated into a single heterozygous virion (G4:G7). (**A**) When this virion subsequently infected a new B cell and template switching occurred during reverse transcription, a recombinant retroviral DNA sequence was generated, and all subsequent progeny virions were of this recombinant genotypes, for example, as was shown on the scheme for putative recombinant forms (G4/G7:G4/G7) and G4/G7/G4: G4/G7/G4. (**B**) The recombination would occur during heterozygous virion formation, already in the first passage (between the two co-infecting RNAs during heterozygous virion formation).

**Table 1 pathogens-13-00178-t001:** Identity and origin of the 27 sequences used as references in the study.

No	GenBank Accession No	Geographic Region	Genotype	Identity Code & Source
1	K02120	Japan	1	Sagata et al. (1985) [19]
2	D00647	Australia	1	Coulston et al. (1990) [20]
3	M35242	USA	1	Mamoun et al. (1990) [21]
4	FJ808597.1	Argentina	2	Rodriguez et al. (2009) [22]
5	AF399704.3	Brazil	2	Camargos et al. (2004) ‡
6	EF065650.1	Japan	3	Zhao et al. (2007) [23]
7	EF065647.1	USA	3	Zhao et al. (2007) [23]
8	K02251.1	Belgium	4	Rice et al. (1984) [24]
9	M35238	France	4	Mamoun et al. (1990) [21]
10	MK820044	China	4	Yang et al. (2019) [25]
11	M35240.1	Belgium	4	Mamoun et al. (1990) [21]
12	FJ808595	Argentina	4	Rodriguez et al. (2009) [22]
13	EF065645.1	Costa Rica	5	Zhao et al. (2007) [23]
14	EF065643.1	Costa Rica	5	Zhao et al. (2007) [23]
15	AY185360.2	Brazil	6	Camargos et al. (2004) ‡
16	MH341523	India	6	Gautam et al. (2018) [26]
17	S83530.1	Italy	7	Molteni et al. (1996) [27]
18	HM563749.3	Russia	7	Rola-Łuszczak (2013) [15]
19	EU262555	Poland	7	Rola-Łuszczak (2013) [15]
20	JQ675759.1	Russia	8	Lomakina et al. (2013) §
21	HM563764	Ukraine	8	Rola-Łuszczak (2013) [15]
22	LC080668	Bolivia	9	Polat et al. (2016) [28]
23	LC080659	Bolivia	9	Polat et al. (2016) [28]
24	KU233547	Thailand	10	Lee et al. (2016) [29]
25	LC154848	Myanmar	10	Moe et al. (2020) [30]
26	OK945975	Kazakhstan	12	Sultanov et al. (2022) [31]
27	OK945977	Kazakhstan	12	Sultanov et al. (2022) [31]

‡ Camargos et al. 2004; direct submission to GenBank. § Lomakina et al. 2013; direct submission to GenBank.

**Table 2 pathogens-13-00178-t002:** Characteristics of the *env* clones of studied samples 1S and 4T.

No	GenBank Accession No	Genotype	Identity Code of Samples
1	JQ353633	7	1S-c6
2	JQ353634	7	1S-c7
3	JQ353635	7	1S-c5
4	JQ353636	7	1S-c8
5	JQ353637	4	1S-c12
6	JQ353638	recombinant 4/7	1S-c2
7	JQ353639	4	1S-c19
8	JQ353640	recombinant 4/7	1S-c9
9	JQ353641	4	1S-c15
10	JQ353642	4	1S-c14
11	JQ353643	4	1S-c13
12	JQ353644	4	1S-c17
13	JQ353645	7	1S-c3
14	JQ353646	recombinant 4/7	1S-c11
15	JQ353647	4	1S-c20
16	JQ353648	4	1S-c18
17	JQ353649	recombinant 4/7	1S-c1
18	JQ353650	4	1S-c10
19	JQ353651	7	1S-c4
20	JQ353652	4	1S-c16
21	JQ353653	7	4T-c15
22	JQ353654	7	4T-c13
23	JQ353655	recombinant 4/7	4T-c19
24	JQ353656	7	4T-c11
25	JQ353657	4	4T-c7
26	JQ353658	recombinant 4/7	4T-c1
27	JQ353659	4	4T-c6
28	JQ353660	4	4T-c3
29	JQ353661	recombinant 4/7	4T-c20
30	JQ353662	7	4T-c12
31	JQ353663	recombinant 4/7	4T-c21
32	JQ353664	7	4T-c17
33	JQ353665	7	4T-c16
34	JQ353666	4	4T-c2
35	JQ353667	7	4T-c18
36	JQ353668	4	4T-c5
37	JQ353669	4	4T-c4
38	JQ353670	4	4T-c8
39	JQ353671	7	4T-c14
40	JQ353672	4	4T-c9

## Data Availability

The data presented in this study are openly available in GeneBank: 2S–5S (JF720350–JF720353); 5T (HM563754); 6T (HM563783); 1C, 7C–10C (JF720354–JF720358); 2Z (JQ320302); 4Z (HM563779) and 5Z (HM563780). The clones of the isolates 1S and 4T were for JQ353633–JQ353652 and JQ353653–JQ353672, respectively.

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
