# Peer review of "Genetic Variability of Bovine Leukemia Virus: Evidence of Dual Infection, Recombination and Quasi-Species"

_pathogens, 2024, doi:10.3390/pathogens13020178_

Round 1

Reviewer 1 Report

Comments and Suggestions for Authors

Dear authors,

thank you very much for this interesting manuscript.

I would like to submit to your attention some suggestions, as following.

·         LINE 79-83: in my opinion, target gene of the described two-step PCR is missing; furthemore, the reference manuscript (n. 16) is not available in the internet. Actually, it is reported in the introduction paragraph (line 62-64) and also in results paragraph (line 110, as expected PCR product), but in my opinion, a remind here sholud be appreciated.

·         LINE 123: “...the presence of 70 and 60 polymorphic sites... “, “of” is missing.

·         LINE 135-139: please, move these lines on the left, in the sheet.

·         LINE 186: “....(1Sc-9)...”, maybe, it would be “1S-c9”.

·         FIGURE 8, caption: line 217, “Branches are shown in violet...”, in my opinion, it is black.

·         LINE 227-229: please, could you explain the distance below 0.3% regarding the 14 isolates? “0.3%” is a sort of cut-off? It seems not coherent with lines 111-113 (“...with pairwise identities ranging from 99.9% - 3 samples- to 100% - 11 samples”).

·         DISCUSSION: could you speculate about a correlation beetween genetic variations and clinical evolution of the disease? Furthemore, there is no evidence of epidemiological data regarding the infected cattle and infected herds, from which the samples originated.

·         LINE 290-294: in my opinion, this part should be moved into the result paragraph; furthemore, the supplementary figure 4 should be better explained and made coherent with the text (for instance, in the caption, dN and dS sites are not reported).

·         LINE 297-300: in my opinion, also the supplementary figure 5 should be better explained and made coherent with the text (for instance, the variations are not correlated to the peaks reported in Supp. Figure 4).

·         LINE 309-314: could you consider the possibility that a recombination event in BLV would occur already in the first passage (heterozygous virion)?

Author Response

Response to Reviewers

Thank you very much for taking the time to review this manuscript. Please find the detailed responses below and the corresponding corrections in track changes in the re-submitted files. In the following pages we have addressed all of the comments from the reviewers. Reviewers’ comments are in bold and authors’ answers in plain text below, prefaced by “AU:”

Reviewer 1

Dear authors, thank you very much for this interesting manuscript. I would like to submit to your attention some suggestions, as following.

AU: The authors thank the Reviewer for their supportive comments.

  1. LINE 79-83: in my opinion, target gene of the described two-step PCR is missing; furthemore, the reference manuscript (n. 16) is not available in the internet. Actually, it is reported in the introduction paragraph (line 62-64) and also in results paragraph (line 110, as expected PCR product), but in my opinion, a remind here sholud be appreciated.

AU: These points are well taken, and we have now revised the text and added Suppl. Table 1 and Suppl. Figure 1 to show detail information on primers sequences for amplification of target gene, env –gp51  (please see lines 80-81 and line 128 in the text).

  1. LINE 123: “...the presence of 70 and 60 polymorphic sites... “, “of” is missing.

AU: Thank you for pointing this linguistic error, it has been corrected on line 140.

  1. LINE 135-139: please, move these lines on the left, in the sheet.

AU: We cannot move these lines to the left, as it was prepared according to the Microsoft Word template based on the Pathogens guidelines.

  1. LINE 186: “....(1Sc-9)...”, maybe, it would be “1S-c9”.

AU: Thank you for pointing this typographical error, it has been corrected on line 213.

  1. FIGURE 8, caption: line 217, “Branches are shown in violet...”, in my opinion, it is black.

AU: Indeed. We have changed our wording in line 321 of the revised manuscript. We have also revised the Figures 7 and 8 and rewritten the legend to the Figures 7 and 8 to describe them more accurately in line 243-252 and in line 292-323.

  1. LINE 227-229: please, could you explain the distance below 0.3% regarding the 14 isolates? “0.3%” is a sort of cut-off? It seems not coherent with lines 111-113 (“...with pairwise identities ranging from 99.9% - 3 samples- to 100% - 11 samples”).

AU: Thank you for this comment.  Yes, the value of 0.3% for the BLV isolates was a threshold initially set for the study and not changed in the final manuscript. Please see revised text on line 346-347.

  1. DISCUSSION: could you speculate about a correlation beetween genetic variations and clinical evolution of the disease? Furthemore, there is no evidence of epidemiological data regarding the infected cattle and infected herds, from which the samples originated.

AU: The authors appreciate the constructive criticism and agree that, based on the results, the range of genetic variation observed for the sequences of 1S and 4T isolates should not be correlated with virus persistence in infected hosts without additional study. However, considering the observed phenomenon, we could not leave it without adding a deductive approach. We have now revised the text and added some new hypotheses (see lines 407-415 of the text).

  1. LINE 290-294: in my opinion, this part should be moved into the result paragraph; furthemore, the supplementary figure 4 should be better explained and made coherent with the text (for instance, in the caption, dN and dS sites are not reported).

AU: Valid point, line 290-294 has been moved to the Results section where positive selection sites are detailed (see line: 156-166). Suppl. Figure 4 has been revised as recommended by both Reviewers. Dn/ds analysis was performed again, separately for the 1S and 4T isolate clones, and the new results are presented in Suppl. Figure 5A and 5B, respectively. The legend below Suppl. Figure 5A-B now provides the necessary explanation of the analysis (peaks of positive selection are identified). In the Materials and methods section, additional information has been added on the calculation of the dn/ds ratio (see line 98-100).

  1. LINE 297-300: in my opinion, also the supplementary figure 5 should be better explained and made coherent with the text (for instance, the variations are not correlated to the peaks reported in Supp. Figure 4).

AU: Thank you for this comment. Suppl. Figure 5 with the amino acid alignment of 40 clones has now been split into Suppl. Figure 6 and Suppl. Figure 7 showing the codon alignment of 20 clones of isolate 1S and 20 clones of isolate 4T, respectively. Both figures have been adapted from the results presented in Suppl. Figure 5A-B and expanded description in the legend. Sliding windows where positive selection is indicated are highlighted in the figures.

  1. LINE 309-314: could you consider the possibility that a recombination event in BLV would occur already in the first passage (heterozygous virion)?

AU: In order to provide a fully comprehensible scheme to illustrate recombination, we proposed to create a heterozygous virion in the first step and introduced the recombination process in the second step (as was shown in Figure 9A). Nevertheless, we generally believe that recombination can occur during reverse transcription, when parts of two co-packed RNAs are used as templates for DNA synthesis. Therefore, we have now added this second possibility as Figure 9B (see Figure 9B, Figure 9 legend on lines 480, 484 and 490-492).

Reviewer 2 Report

Comments and Suggestions for Authors

In this manuscript, Aneta Pluta et al. report phylogenetic analysis of bovine leukemia virus sequences isolated from cattle in the Western Siberia -Tyumen and South Ural-Chelyabinsk. The study indicated the genotypes 4 and 7 are spread in these areas. They detected dual infection of genotype 4 and 7 sequences in two cattle. In these cattle, they identified 8 breakingpoints of recombination between two genotypes. The study is of interest for both BLV epidemiology and retrovirology. However, there are some points that deserve clarification, especially regarding the biological relevance of the observations and the novelty of some of the findings in BLV-infected cattle.

Main points

1.     The authors applied common method to analyze proviral sequences of BLV that PCR amplify the region of interest followed by cloning and sequencing of the PCR products. However, this can be problematic especialy in this study when sequences are amplified from a multi-template population as artificial chimeras can be generated. The formation of chimeras by PCR leads to the genetic mixing of sequences which can generate new diversity that is not present in the initial sample(Smyth et al., gene 2010 PMID: 20833233). The authors should additionally perform a highly sensitive PCR methods to confirm recombination between genotype 4 and 7 such as directrly amplify genomic DNA with genotype specific primers, primers targeting breaking point of recombination or inverse PCR of proviral genome.

2.     The authors define quasispecies as the population of viruses that inhabit a single host. However, only proviral genome sequences were analyzed. Proviral genome sequences could be mutated with several APOBEC3 proteins after integration, indicating not reflecting genuine viral genome sequences. The authors should analyze virus RNA sequences for analysis and discussion of quasispecies of BLV.

3.     Line 285: The authors analyzed positive selection of 40 sequences of two genotypes from two hosts altogether. To analyze virus evolution within host, positive selection analysis should be done individually in each host. Moreover, the authors assumed that two genotypes within a host exist not by evolution but by dual infection. Therefore, two genotypes should be also analyzed separately.

4.     Line 277: In this study, the dual genotypes infected cattle indicated very low proviral load such as 0.035 % and 1.13 % of PBMCs were infected with BLV. Generally, dual infection likely to occur under high proviral load because BLV infection and genome integration occur by chance. Please describe the probable reason in discussion section why such rare events observed in this study.

Minor points

1.     Please describe detailed information of primers, fidelity, and error rate of DNA polymerase which authors used for 2-step PCR.

2.     Phylogenetic analysis of not only env sequences but also other virus sequences such as LTR, tax, gag, and pol would be informative.

Author Response

Response to Reviewers

Thank you very much for taking the time to review this manuscript. Please find the detailed responses below and the corresponding corrections in track changes in the re-submitted files. In the following pages we have addressed all of the comments from the reviewers. Reviewers’ comments are in bold and authors’ answers in plain text below, prefaced by “AU:”

Reviewer 2

Comments and Suggestions for Authors

In this manuscript, Aneta Pluta et al. report phylogenetic analysis of bovine leukemia virus sequences isolated from cattle in the Western Siberia -Tyumen and South Ural-Chelyabinsk. The study indicated the genotypes 4 and 7 are spread in these areas. They detected dual infection of genotype 4 and 7 sequences in two cattle. In these cattle, they identified 8 breakingpoints of recombination between two genotypes. The study is of interest for both BLV epidemiology and retrovirology. However, there are some points that deserve clarification, especially regarding the biological relevance of the observations and the novelty of some of the findings in BLV-infected cattle.

AU: We thank the Reviewer for the careful and insightful review of our manuscript. We address all of the concerns of the Reviewer here.  

Main points

  1. The authors applied common method to analyze proviral sequences of BLV that PCR amplify the region of interest followed by cloning and sequencing of the PCR products. However, this can be problematic especialy in this study when sequences are amplified from a multi-template population as artificial chimeras can be generated. The formation of chimeras by PCR leads to the genetic mixing of sequences which can generate new diversity that is not present in the initial sample (Smyth et al., gene 2010 PMID: 20833233).

AU: Thank you for this comments. While the authors are aware that during subsequent PCR cycles, partially elongated products may bind to heterologous templates, acting as primers that can be extended to form artificial chimeras, our PCR cycling conditions were determined rationally: (1) we used an incubation time of 2 min to be sufficient to anneal the primers, (2) the number of cycles did not exceed 35, (3) PCR reagents were selected using a rational design approach, (4) primer concentrations, 0. 5 μM were used such that any partially extended primers were competed for the template for binding sites by non-extended primers in subsequent cycles, (5) a homogeneous band was observed on the agarose gel, (6) two-step PCR using Thermo Scientific DyNAzyme II DNA polymerase was tested on other population DNA samples with a double template and, as a result, no formation of artificial chimeras was observed during PCR, what was confirmed by sequencing. Therefore, we believe that our PCR protocol is appropriate and accurately reflects the diversity of the population of 1S and 4T isolates, rather than artifacts introduced by the process itself. We have now added new results assessing the possibility of creating artificial recombinants by PCR reactions in lines 329-341.

The authors should additionally perform a highly sensitive PCR methods to confirm recombination between genotype 4 and 7 such as directrly amplify genomic DNA with genotype specific primers, primers targeting breaking point of recombination or inverse PCR of proviral genome.

 AU: Thank you. Molecular analysis of BLV field isolates shows a high degree of homology between BLV proviral sequences compared to the genetic variation found within the other genera within the family Retroviridae. The authors ensure that the two-step PCR used in the study is the most sensitive method for analyzing provirus sequence variations in weak-positive DNA samples from BLV-infected cattle and this method also allowed the detection of recombination between G4 and G7 genotypes (the literature has not yet provided another more appropriate method for this purpose).

The 5' (forward) and 3' (reverse) primers for the amplification of env-gp51 are located in conserved regions of the provirus genome, at pol (primer P4796- 98.7% identity; P4833 – 99.8% based on 320 sequences from Genbank) and env-gp30 (primer P5791- 99.2%; P5734 – 98.3% - based on 320 sequences available in Genbank), respectively (as shown in the newly added Suppl. Figure 1). The authors are convinced that there is no need to design primers that are genotype-specific, as the differences between particular genotypes are not significant enough, and are caused by the presence of point mutations randomly scattered.

However, in order to fulfil the Reviewer’s concerns and expectations, the Authors designated  two primers, at the most variable of all possible sites, located at the recombination sites (File 1, please see recombinant  1S-c1 representing 10240bp- G7, 241-541bp- G4 and 542-804bp- G7).

Then the PCR reaction was optimized and performed with different clones representing both the G4 and G7 genotype and with clones representing all identified recombinants; resulting in equal bands for each amplification reaction (see in File 2). The positive results for all samples tested only confirmed that genotype-specific primers and primers targeting the breaking point of recombination, which differs by 3-4 nucleotides, are not an effective tool as a means of confirming recombination.

Unfortunately, the inverse PCR of proviral DNA was impossible to perform from these two DNA samples, because, as with NGS, DNA samples with provirus present below 500 copies per 105 cells IPCR is of limited use, so in this case the technology was not used due to the provirus load in the samples being too low (on lines 510 and 515). The authors believe that, despite these limitations in performing additional PCRs, the present results shed some new light on BLV variability and may serve as a good starting point for further research.

  1. The authors define quasispecies as the population of viruses that inhabit a single host. However, only proviral genome sequences were analyzed. Proviral genome sequences could be mutated with several APOBEC3 proteins after integration, indicating not reflecting genuine viral genome sequences. The authors should analyze virus RNA sequences for analysis and discussion of quasispecies of BLV.

AU: To date, molecular analysis of BLV field isolates has been performed on proviral genome sequences. Our results are the first to report the presence in two animals, 1S and 4T, two clearly distinct proviral populations (intrastrain variability) showing quasispecies nature (please see line 381-389) based on proviral sequences. The authors agree with the Reviewer that they should also analyse the RNA sequences of the virus to confirm the quasispecies, however, extraction of BLV RNA from naturally infected animals is not possible due to the specific nature of their replication cycle. The life cycle of BLV is different from HIV. BLV replicates mainly by polyclonal expansion of infected B lymphocytes. The proviral DNA preferentially integrates into actively transcribed regions of the host genome, which allows active viral replication and the production of progeny virions during the early period of infection, which usually lasts from 2 to 4 weeks after infection. This is followed by a humoral host immune response against the viral glycoprotein gp51 and the capsid protein p24. Almost simultaneously with the seroconversion period, cytotoxic T-lymphocytes with anti-gp51 and anti-Tax specificity and CD4+ helper T-lymphocytes also appear, contributing to a rapid increase in the total number of CD8+ lymphocytes. This results in the clonal expansion of latently infected B lymphocytes, which do not express viral proteins on the cell surface, becoming the main mechanism of viral spread at a later stage of infection. This is a typical latency state for BLV, confirmed by the lack of detection of antigenic protein expression by techniques such as ELISA, immunoprecipitation or Western blot. Some researchers report that when they used flow cytometric sorting of peripheral blood cells and performed an amplification reaction using a sensitive RT-PCR technique, it showed tax/rex mRNA expression in single B lymphocytes, which was accompanied by persistent lymphocytosis. However, the cattle in which dual infection and quasispecies were detected did not have persistent lymphocytosis.

The mutations observed in all sequences of the isolates were not random but common for to the G4 and G7 genotypes. BLV integrates less frequently as compared to HIV, therefore the function and importance of APOBEC3 in BLV is a secondary issue, with minimal impact on mutation formation. If the mutations were indeed caused by APOBEC3, we would also observe STOP codons within the sequences. However, neither for the sequences of the 14 isolates nor for the 2 isolates from the dual infection obtained from PCR products or clones, respectively, did we observe STOP codons. As well known APOBEC proteins are cytosine deaminases that can deaminate cytosine to uracil (C-to-U) in single-stranded nucleic acids. In our study we did not observe a significant accumulation of this type of mutations.

Some co-authors claim that it is no matter whether the mutation is caused by an error in viral replication or by post-integration modification by APOBEC3 into viral diversity. This is interesting angle that the literature has not yet provided concrete answers to.

  1. Line 285: The authors analyzed positive selection of 40 sequences of two genotypes from two hosts altogether. To analyze virus evolution within host, positive selection analysis should be done individually in each host. Moreover, the authors assumed that two genotypes within a host exist not by evolution but by dual infection. Therefore, two genotypes should be also analyzed separately.

AU: We have addressed this point in response #8 above to Reviewer 1. An valid point, however, given that the aim of this analysis was to determine BLV variation in a DNA sample from a single animal, we prefer not to split the calculation for the two genotypes within a single host.

  1. Line 277: In this study, the dual genotypes infected cattle indicated very low proviral load such as 0.035 % and 1.13 % of PBMCs were infected with BLV. Generally, dual infection likely to occur under high proviral load because BLV infection and genome integration occur by chance. Please describe the probable reason in discussion section why such rare events observed in this study.

AU: To explain this inconsistency, the authors assumed that virus transmission could have occurred almost simultaneously through genotypes 4 and 7 from two animals with high proviral loads, cow A with G4 and cow B with G7, leading to a double infection in cow C. We have already discussed this topic of co-infection rather than super-infection in lines 361-375 and additionally considered this event as coincidental in line 376 and line 520-521.

Cows A and B were probably highly positive, as such cases have also been reported in these herds. We also speculate that another possible mode of virus transmission to cow (C) was most likely blood sucking midges breeding around the edge of water of the Koktyul and Koylega rivers and small lakes in the West Siberian Plain. If this was the case, it means that the host infected with the recombinant virus was almost simultaneously infected with genotypes 4 and 7 or that it received a double infection from the source. We have now added another possible route for this transmission and inserted references in lines 376-380.

Minor points

  1. Please describe detailed information of primers, fidelity, and error rate of DNA polymerase which authors used for 2-step PCR.

AU: We have addressed the primers in response #1 above to Reviewer 1.

Thermo Scientific DyNAzyme II DNA Polymerase, used for nested PCR, is recombinant thermostable DNA polymerase with an improved thermal stability at high temperatures. The polymerase mutation rate is 2.28 x 10-5 mutations per base pair per template duplication. After 35 cycles of PCR amplifying a 804 bp template, 64,16% of the product DNA molecules may have contained 1 (nucleotide) error each. This means that 35,8% of the product molecules may have been entirely error-free. Considering two PCR reactions of 35 cycles each performed with this polymerase, every product molecule contained an averaged 1.28 errors. The next calculation is as follows: 20 clones 1S (or 20 clones 4T) x 1.28 errors = 25.6 errors (nt); 804 bp x 20 clones = 16080nt; 16080nt = 100% identity; 25.6 nt = 0.1592%. Mean genetic distance among 20 clones was 2.0% (range: 0.1% - 4.5%) and 2.1% (range: 0% - 4.7%) for isolates 1S and 4T. Based on the above polymerase fidelity calculations, we concluded that this polymerase was suitable for studying the variation of the BLV isolates described in this study (please see line 87-91, and the Supl. Table 1).

In order to estimate the potential contribution of PCR errors to sequence variability, three independent PCR products for each DNA sample were sequenced (as was reported in line 91-93).

  1. Phylogenetic analysis of not only env sequences but also other virus sequences such as LTR, tax, gag, and pol would be informative.

AU: Characterization of variation in the coding region of the viral envelope protein gp51 was the main objective of the study. This protein contains a number of important antigenic determinants that are associated with the induction of a strong humoral and cellular immune response at the first stage of infection. Mutations in these domains may be important for virus. Our previous studies were based on the analysis of a 444 bp fragment of the env gene. In the current study, we estimated the levels of nucleotide diversity of the env gene fragment encoding the entire env gene glycoprotein gp51 from BLV-infected cattle. In this study, we did not include regions of the BLV genome, such as LTR, tax, gag and pol, because they are highly conserved and phylogenetic analysis is not possible, or they do not contain or have not so far identified important domains related to host immune defence in these regions, or they do not have mutations specific to the G4 and G7 genotypes described in the literature.